# Who is (not) complying with the U. S. social distancing directive and why? Testing a general framework of compliance with virtual measures of social distancing

**Russell H. Fazio** [1]\*, **Benjamin C. Ruisch**[2], **Courtney A. Moore**[1], **Javier A. Granados Samayoa**[1], **Shelby T. Boggs**[1], **Jesse T. Ladanyi**[1]

1 Department of Psychology, The Ohio State University, Columbus, Ohio, United States of America,
2 Institute of Psychology, Leiden University, Leiden, The Netherlands

\* fazio.11@osu.edu.

**Data Availability Statement:** The datafile is available on the Open Science Framework at https://osf.io/359et/.

## Abstract

A study involving over 2000 online participants (US residents) tested a general framework regarding compliance with a directive in the context of the COVID-19 pandemic. The study featured not only a self-report measure of social distancing but also virtual behavior measures—simulations that presented participants with graphical depictions mirroring multiple real-world scenarios and asked them to position themselves in relation to others in the scene. The conceptual framework highlights three essential components of a directive: (1) the source, some entity is advocating for a behavioral change; (2) the surrounding context, the directive is in response to some challenge; and (3) the target, the persons to whom the directive is addressed. Belief systems relevant to each of these three components are predicted, and were found, to relate to compliance with the social distancing directive. The implications of the findings for public service campaigns encouraging people to engage in social distancing are discussed.

## Introduction

Until a vaccine has been disseminated widely, minimizing the spread of COVID-19 requires that people change their behavior. People are urged to wash their hands frequently, use hand sanitizer, disinfect surfaces, and wear masks. Above all, since mid-March 2020 when the pandemic reached a critical level in the United States, government leaders and health experts have pleaded with citizens to engage in social distancing–that is, to deliberately increase the physical space between themselves and other people. The mantra "stay six feet away from others" has been repeated regularly. Most states took even stronger action, imposing "shelter-in-place" orders for weeks, sometimes months, in the interest of minimizing contact among people. Even after dire economic concerns prompted some states to gradually begin the process of re-opening, the plea to engage in social distancing has, if anything, been emphasized all the more. Tape on the floors of stores designates intervals of six feet; restaurants and bars are required to

**Funding:** This work was supported by a RAPID grant from the National Science Foundation under Award ID BCS-2031097 (RHF). The funders had no role in study design, data collection and analysis, decision to publish, or preparation of the manuscript.

**Competing interests:** The authors have declared that no competing interests exist.

situate tables six feet apart; public parks are outlined with circles demarcating six feet of separation.

Rarely has the entire population been called upon to exhibit immediate behavior change in compliance with an urgent directive. That raises an important question: who is or is not complying? Understanding who chooses to practice (or not) social distancing–and why–is crucial for the design of effective public service campaigns, both now, and during the occurrence of future pandemics. Whom should such campaigns target? What specific beliefs should be addressed?

Theory and research concerning compliance, i.e., behavioral change in response to an explicit or implicit request, is central to social psychology. As delineated in Kelman's classic treatise regarding the nature of the changes that may be fostered by a communication, any such behavioral change may range from passive conformity with a source's message in the interest of avoiding disapproval to a more internalized, private acceptance of the inherent value of the message [1]. Although internalization promotes more generalized behavior change that is not dependent upon normative approval or disapproval, the key for the present purposes, and how we define compliance here, is whether a given individual responds to the request by *engaging in the desired behavior*, whatever the reason may be. The field has acquired substantial knowledge regarding social influence tactics that promote compliance [2], including such classic approaches as the foot-in-the-door [3], door-in-the-face [4], and low-balling [5]. In addition, the impact of both descriptive and injunctive norms, and the interplay between them, has been examined extensively [6, 7].

However, as some scholars have noted [8], the field lacks a general theoretical framework about who is likely to comply with a directive, and why they might or might not. Such a framework is particularly important when considering a directive calling for compliance and behavior change on a large scale, as is currently true of the social distancing directive. The major aim of the current research is to test such a theoretical framework regarding the who and why of compliance.

## Compliance with a directive: What's involved?

Any directive is open to interpretation and ultimately will be assessed as warranting or not warranting compliance. Only if deemed justified on the basis of one's reasoning regarding the merits of the source's arguments, or on the basis of one's mere respect for the source, is the directive likely to promote behavioral change. Yet, one of the core principles of social psychology is that individuals construct their own reality [9–11]. Such constructions influence and are influenced by the information to which individuals choose to expose themselves [12], their exploratory behavior, and ultimately the accuracy of their understanding of reality [13, 14]. Thus, any given directive will be viewed through the lens of the individual's knowledge, beliefs, and attitudes. Decades of research demonstrate the pervasive influence of such factors on judgments and decisions [15–17]. An excellent example of such processes, and their potential significance within the domain of health, stems from issues that are now prominent regarding parental vaccination of children. Endorsement of such vaccine conspiracy beliefs as "pharmaceutical companies cover-up the dangers of vaccines" is strongly associated with parents not complying with routine vaccination recommendations for their children [18, 19].

Such considerations raise the question of what might be regarded as the essential components of a directive. Our theoretical framework highlights three: (1) the *source*: the entity advocating for behavioral change; (2) the surrounding *context*: the challenge the directive addresses; and (3) the *target*: the persons to whom the directive is addressed. Critically, the framework guiding the present research and our selection of predictor variables contends that

belief systems relevant to each of these three components will influence the likelihood of compliance. Is the source to be trusted? What does the surrounding context imply about the seriousness of the challenge? Are there individual propensities that affect responsivity to the directive? Thus, a complex network of beliefs will affect who chooses to comply or not, and for what reasons. Some individuals' belief systems will lead them to assess the directive favorably, thus increasing the likelihood of behavior change. Others will reach less positive conclusions and, hence, are likely to fail to respond appropriately.

This conceptual framework bears some similarity to the classic "who says what to whom" question pursued by Carl Hovland and his associates in the Yale Communication Group regarding persuasive communication [20]. Although the emphasis on source variables ("who") is indeed parallel with our framework, our focus does not include a consideration of any specific message variables ("what"), largely because any directive concerning a large-scale challenge, like the social distancing directive in response to the COVID-19 pandemic, is likely to involve diverse messages delivered across multiple media. Our interest is in how individuals respond to the general plea, not any specific persuasive message in service of that plea. Finally, whereas the Yale Communication group was largely concerned with personality variables that might relate to general persuasibility ("whom"), our framework's focus on target characteristics is more specific. The concern is with characteristics that are likely to relate to receptivity to a given directive in light of a given challenge.

## Measuring social distancing

In examining compliance, the challenge rests in how to assess social distancing. Observation of individuals' behavior in the field, arguably the "gold standard" in research on behavioral compliance, followed by extensive interviewing of those who are and are not maintaining social distance, is simply impractical. For this reason, the field's dominant approach is to ask people to report the frequency with which they socially distance. However, the problems associated with self-reports of behavior have been discussed for decades. Individuals may over-report their social distancing to convey a socially desirable impression to others and themselves [21–24]. Moreover, self-reports may be all the more problematic to the extent that they rely on retrospective memory regarding past behavior [25, 26]. Even more troubling, some of the very characteristics and beliefs we predict will affect responsiveness to the directive may also influence how a person (mis)represents their social distancing on self-report measures. For example, strongly valuing one's identity as a liberal or as a conservative might promote the reconstruction of memory regarding one's social distancing behavior in the very direction implied by that valued identity. Likewise, believing oneself to be a compassionate person who is concerned about others' vulnerability to COVID-19 may promote an exaggerated reconstruction of the extent to which one is practicing social distancing.

We thus supplemented self-reports of social distancing with a more innovative, behaviorally-oriented approach. We simulate social distancing behavior with graphical depictions mirroring real-world scenarios. These involved a variety of situations in which individuals commonly encounter other people (e.g., sidewalks, a crosswalk, park pathways, a plaza, a grocery, a beach, a library, and a coffeeshop) and, hence, experience an opportunity to engage in social distancing. In each case, we asked participants to position themselves in relation to others in the scene. Hence, the virtual social distancing scenarios required a concrete, "in-the-moment" behavioral decision, which could vary in the degree to which participants did or did not distance themselves from others. For example, in one scenario participants chose whether to cross a park via a circuitous but isolated path versus a more-direct but crowded route. In another, they were presented an aerial image of a crowded beach and asked to click on the spot

where they personally would lay down their towel. Yet another presented an interactive image of two people approaching each other in a crosswalk for which participants were asked to move a slider that shifted the walkers from the center of the crosswalk to the distance that they personally would leave between themselves and the other individual.

Our argument regarding the value of these virtual behavior scenarios parallels a relevant empirically-supported proposition regarding attitudes as predictors of behavior. Attitude measures are more likely to predict behavior when they match the behavior in terms of specificity regarding the action in question and the context in which the action is performed [27, 28]. Similarly, the simulated scenarios closely match real-life situations in terms of their concreteness. They offer a means, in addition to a self-report, of indexing the extent to which individuals make decisions that accord with the principle of social distancing.

The validity of the self-report measure of social distancing and, more importantly, the novel virtual behavior measure has been established by recent longitudinal data [29]. Four months after participating in studies involving these measures, over 2000 participants indicated whether they had contracted COVID-19 during the interim period. Both measures proved predictive. However, illustrating both the problems associated with self-reports and the value of the novel measurement approach, the virtual behavior measure accounted for unique variance when the two predictors were considered simultaneously, whereas the self-report measure did not.

## Predictor variables

This study examines the relation between social distancing and various predictor variables. Surely, innumerable variables may merit consideration for inclusion in such research. In our case, the selection of predictors was guided by our framework regarding the essential components of a directive. How trustworthy are those espousing the plea? How serious is the challenge that the requested behavioral change is intended to address? Are there particular individual characteristics that are likely to influence receptivity? Hence, to test our compliance framework, the predictors involve our three classes of beliefs–those regarding the source of the directive, the surrounding context posed by the challenge, and additional characteristic of the targets themselves.

**Beliefs about the source.**   The primary source of the social-distancing directive is government and health officials. The latter are medical scientists or liaisons representing the scientific community. Given the distinction in the literature between valuing science as means of acquiring knowledge and trusting scientists and their work [30], we hypothesized that both (a) greater belief in science and (b) greater trust in scientists would relate positively to compliance.

Given the highly polarized sociopolitical context and the politicization of the pandemic within the United States, assessing faith in government officials is more complex. The messages the public received from various officials were not consistent, with President Trump often downplaying the seriousness of the pandemic relative to what state Governors and health experts were communicating early in the pandemic while most states were under "shelter-in-place" orders [31–33]. Accordingly, we separately assessed trust in the President's and Governors' leadership regarding the pandemic, predicting that these measures may relate differently to compliance.

**Beliefs about the context.**   In accord with our guiding framework, we generated a series of items related to the challenge that the social distancing directive aimed to address. These involved assessments of the seriousness of the pandemic and support for social distancing. They were included to examine the hypothesis that greater concern about the virus and

positive attitudes toward the directive would be associated with more social distancing. We tested a similar hypothesis regarding accurate knowledge about COVID-19 by administering a brief quiz about the virus. More knowledgeable individuals were expected to display more distancing.

**Target characteristics.**    Two sets of target characteristics were expected to relate to individuals' receptivity to the plea to engage in social distancing: (a) beliefs relevant to disease or views of the government and (b) more general characteristics of the individual relevant to the plea to socially distance. Perceived vulnerability to disease [34] and its concomitant disgust sensitivity [35] were expected to relate positively to social distancing. Similarly, general compassion [36] and concern for others' vulnerability to the coronavirus were expected to predict distancing. Our conceptual reasoning led us to identify two additional receptivity-related beliefs that would affect individuals' acceptance of the social distancing directive as a result of the influence that they would exert on both beliefs about the source and beliefs about the challenge. First, we expected political conservatism to relate to less social distancing. Our reasoning was that more conservative individuals traditionally place greater emphasis on economic matters, and social distancing directives may be viewed as a threat to the economy. Moreover, President Trump, a key conservative leader, expressed both an equivocal stance regarding the severity of the pandemic and urgency regarding reopening the economy [32]. The second receptivity-related belief on which we focused was the general tendency to endorse conspiracy theories. A considerable literature points to the significance of conspiratorial ideation as a factor associated with the rejection of scientific findings and recommendations [37–39]. We predicted that such beliefs would promote minimization of COVID-19's severity, and hence relate to less compliance.

Turning to the second set of target characteristics, we also predicted that individual differences in scientific literacy [40] would likely relate to both trust in health experts and the development of accurate knowledge regarding the coronavirus. Hence, scientific literacy was expected to be associated with more distancing. Additionally, a considerable literature highlights the importance of the particular news sources that individuals follow [41–43]. We expected that reliance on more conservative news sources would relate to minimizing the threat posed by the pandemic and less distancing behavior.

## Materials and methods

We recruited a sample from Mechanical Turk. Although not representative of the U.S. population, MTurk samples are considerably more diverse than the student samples used in most psychological research [44, 45], and they perform similarly to non-MTurk samples across many tasks and measures [46, 47], including surveys on political attitudes [48]. Further, our aim is not to make claims regarding the absolute frequency of beliefs and behaviors in the population, but rather to understand how the psychological variables of interest relate to social distancing behavior. Given these aims, we judged the MTurk sample as appropriate for testing our hypotheses. As will become evident, both the very systematic nature of the data and their replication of some relations previously established in the literature attest further to the appropriateness of the MTurk sample.

Past experience with MTurk participants led us to believe that they prefer, and respond most conscientiously, when a study is relatively short. Hence, we included only our social distancing measures, survey items assessing beliefs and knowledge about the pandemic, and various demographics as the elements of a common survey that was completed by all the participants. Subsets of our other predictors were included in four distinct surveys to which participants were randomly assigned. The four subsets involved: (a) source beliefs and science

literacy, (b) news sources and belief in conspiracy theories, (c) compassion and concern for others vulnerability to COVID-19, and (d) perceived vulnerability to disease and disgust sensitivity. Demographic data regarding the participants in each of the four sub-studies are presented in S2 Material; these attest to the comparability of the four randomly-assigned samples.

## Participants

We aimed for sample sizes that would clearly be large enough to obtain stable estimates of the relations with social distancing within each of our four sub-studies [49]. A total of 2,001 MTurk workers (US residents) participated in the common survey (903 women, 1,084 men, 14 no response; $M_{age}$ = 38.66, $SD_{age}$ = 12.33), with about 500 being randomly assigned to each sub-study. They completed the study on May 7–8, 2020, at which time some states had begun to re-open their economies.

## Measures

Ohio State University's Institutional Review Board approved all study procedures (IRB: 2020B0129). After providing informed consent, participants completed the behavioral measures of social distancing, followed by questions regarding the pandemic, the test of COVID-19 knowledge, the unique set of predictor variables for the study to which the participant had been randomly assigned, and finally a series of demographic questions. All of the measures and the datafile are available at https://osf.io/359et/.

**Virtual social distancing behaviors.** Ten graphical scenarios comprised the virtual measure of social distancing behavior. Examples include: (a) An image of two people approaching each other in a crosswalk. Participants moved a slider that shifted the walkers from the center of the crosswalk to the distance that they would prefer. (b) An aerial image of a crowded plaza that participants were asked to traverse by drawing a path from a start point located at the southwest end of the plaza to an end point at the northeast end. The length of the paths that participants drew (in pixels) was measured as the data of interest. (c) A graphic depicting a park for which participants used a 4-point scale to indicate whether would they definitely or probably walk via one of two paths. One path was less direct, but also more isolated relative to the many people situated on either side of the alternative path. Still images of these three graphical scenarios are presented in Fig 1. All ten of the behavioral scenarios are described in S1 Material and can be viewed at our demonstration website, http://psychvault.org/social-distancing/. After standardizing scores from each measure, we computed the average as our index of social distancing behaviors (α = .82).

**Predictor variables.** *Questions regarding the pandemic*. The behavioral scenarios were followed by the common portion of the survey, including the self-report measure of social distancing: "Generally speaking, how strictly have you personally been following the "social distancing" recommendations?" to which participants responded on a 7-point scale ranging from "not at all" to "very strictly." They also responded to a number of questions regarding

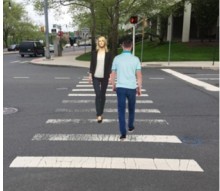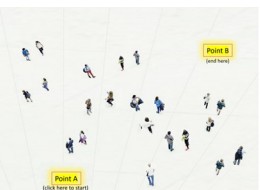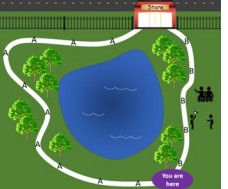

**Fig 1. Example virtual behavior items.**

perceptions of the pandemic. Participants were asked how worried they were about contracting the virus, how likely they were to do so, and how concerned they were about the spread of the virus. They also indicated whether they felt the threat of COVID-19 had been "greatly exaggerated," "somewhat exaggerated," "adequately conveyed," or "not conveyed strongly enough." Yet another item inquired about the tradeoff between economic considerations and safety by asking participants to endorse one of six statements ranging from "Authorities should ONLY focus on protecting people from COVID-19 / the coronavirus, regardless of how much the economy will suffer" to "Authorities should ONLY focus on protecting the economy, regardless of how many people will suffer from COVID-19 / the coronavirus." A more general attitudinal question asked participants to use a 7-point scale to indicate the extent to which they supported or opposed the guideline to engage in social distancing.

*COVID-19 knowledge.* Participants indicated whether each of 13 statements regarding the coronavirus (facts and myths addressed by the Centers for Disease Control and Prevention and the World Health Organization) were true or false. Included were false items such as "Antibiotics are an effective treatment for COVID-19 / the coronavirus" and true items such as "Some individuals who have COVID-19 / the coronavirus do not show any symptoms." The number answered correctly served as our index of COVID-19 knowledge ($\alpha$ = .83).

*Faith in government.* To assess people's trust in different elements of the government, we used four single-item measures all of which involved responding on a 7-point scale ranging from "Not at all" to "Very Much." Specifically, participants were asked to rate the extent to which they "trust President Trump to lead us effectively through the current COVID-19 crisis" and separately whether they trust state governors to do so. They also indicated the extent of their general confidence in President Trump and general confidence that the federal government will address the nation's problems effectively.

*Belief in the value of science.* To assess the extent to which individuals believe in the value of science as the best way to accumulate knowledge about the world, participants responded to a shortened version (the six items with the highest factor loadings) of a scale developed by Farias, Newheiser, Kahane, & de Toledo [50]. Participants rated the degree to which they endorsed statements such as "Science is the most efficient means of attaining truth" on a six-point scale ranging from 1 (Strongly disagree) to 6 (Strongly agree). The average score across the six items was computed as the relevant index ($\alpha$ = .92).

*Trust in scientists.* This variable was assessed using a shortened version (the 11 items, out of 21, with the highest corrected item-total correlations) of the scale developed by Nadelson et al. [30]. Participants rated on a scale ranging from 1 (Strongly disagree) to 5 (Strongly agree) their level of agreement with statements such as "We should trust the work of scientists" and "Scientists ignore evidence that contradicts their work" (reverse coded). The average rating across the eleven items was computed ($\alpha$ = .80).

*Science literacy.* Participants' understanding of basic scientific ideas was assessed using the Civic Scientific Literacy Scale [40]. This scale consists of 11 claims such as "Light travels faster than sound" and "Electrons are smaller than atoms" for which participants indicate agreement or disagreement. The number of correct responses was computed ($\alpha$ = .59).

*Conspiracy theories.* Participants completed the Generic Conspiracist Beliefs scale [37]. The scale consists of 15 items that address a variety of generic conspiracy theories, including "Evidence of alien contact is being concealed from the public," "A small, secret group of people is responsible for making all major world decisions, such as going to war," and "Experiments involving new drugs or technologies are routinely carried out on the public without their knowledge or consent." Participants responded to each statement on a scale of 1 (definitely not true) to 5 (definitely true), with the average rating serving as the measure of general belief in conspiracy theories ($\alpha$ = .96).

*News sources*. Participants were presented with a list of potential News sources: CNN, Fox News, MSNBC, NPR, national newspapers and magazines, social media, and ABC, CBS, or NBC News, as well as the option "do not follow the news." They were asked to select all the sources from which they got their news in the past week. Any who selected an option other than not following the news were then asked to select which one of the outlets they consider to be their primary source of news. Our interest was especially in exposure to Fox News, an outlet whose political leaning is known to be conservative [43].

*Compassion*. To assess general compassion for others, we employed a subset of items of the Interpersonal Reactivity Index [36]. Specifically, we included the 14 items of the scale related to empathic concern (e.g., "When I see someone being taken advantage of, I feel kind of protective towards them") and perspective taking (e.g., "Before criticizing somebody, I try to imagine how I would feel if I were in their place"). Participants responded on a 5-point scale ranging from "Does not describe me well" to "Describes me very well" and their average rating served as the measure of interest ($\alpha = .87$).

*Concern for others' vulnerability to COVID-19*. Four items assessed the extent to which participants experienced empathic concern for people who had contracted COVID-19 or were vulnerable to do so. Participants rated their agreement on a 6-point scale with such statements as "I feel it is my personal responsibility to keep others safe from COVID-19 coronavirus." The average rating was computed ($\alpha = .67$).

*Perceived vulnerability to disease*. Individuals' perceptions of their likelihood of contracting a disease or illness was assessed with the 15-item scale developed by Duncan, Schaller, & Park [34]. Participants rated the degree to which they agreed with statements such as "If an illness is 'going around' I will catch it" on a 5-point scale. After the required reverse-coding of some items, the average response to the 15 scale items was computed ($\alpha = .73$).

*Disgust sensitivity*. The contamination subscale (five items) from the Disgust Scale Revised [35] was used to assess individuals' sensitivity regarding situations that have the potential for the transmission of pathogens. Participants rated on 5-point scales how disgusted they would be by various scenarios such as "A friend offers you a piece of chocolate shaped like dog doo," as well as their agreement with statements such as "I probably would not go to my favorite restaurant if I found out the cook had a cold." The average response to the five scale items served as the measure of disgust sensitivity ($\alpha = .70$).

*Demographics*. In addition to a number of demographic questions (e.g., age, gender, and employment status), participants were asked to identify their political orientation on a scale ranging from 1 (Extremely liberal) to 7 (Extremely conservative).

*Attention check*. The survey concluded with a brief attention check in which participants were informed that a man had seen a beautiful butterfly, and were then asked to select what he had seen: a girl, a day, a fruit, or an insect. Ninety-one percent correctly chose insect. To provide a more conservative test of our hypotheses, we did not exclude participants who failed this attention check. However, none of our conclusions or statistical results are altered to any meaningful degree if these participants are excluded from analyses.

## Results

### Social distancing

To test each of the hypotheses, we examined the multiple correlation between a given variable and our two indices of social distancing–the behavioral and the self-report measures. Table 1 presents the regression data for each of our hypothesized predictor variables. Table 2 displays the correlations among the variables. Turning first to the source, both *belief in the value of science* and *trust in scientists* correlated positively with social distancing. As expected, these

**Table 1. Multiple correlations with virtual behavioral and self-reported social distancing.**

| | R | F | df | Standardized Beta[a] | |
|---|---|---|---|---|---|
| | | | | Behavioral | Self-Report |
| **Beliefs about the Source** | | | | | |
| Belief in Value of Science | .247 | 15.993*** | 494 | .117* | .172*** |
| Trust in Scientists | .372 | 39.754*** | 494 | .299*** | .127** |
| Trust President Trump re COVID-19 crisis | .294 | 23.337*** | 493 | -.309*** | .038 |
| Trust State Governors re COVID-19 crisis | .268 | 19.021*** | 493 | .027 | .255*** |
| Confidence in Federal Gov't Effectiveness | .205 | 10.832*** | 493 | -.223*** | .139** |
| General Confidence in President Trump | .281 | 21.118*** | 493 | -.294*** | .033 |
| **Beliefs about the Context** | | | | | |
| Support Social Distancing Guideline | .650 | 728.207*** | 1995 | .248*** | .498*** |
| Worry about Contracting Virus | .303 | 101.249*** | 1998 | .145*** | .208*** |
| Likely to Contract Virus | .096 | 9.327*** | 1997 | .026 | .081** |
| Concerned about the Spread of the Virus | .409 | 200.380*** | 1997 | .221*** | .257*** |
| Threat (not) exaggerated | .458 | 265.539*** | 1998 | .343*** | .185*** |
| Economy More Important than Safety | .377 | 165.181*** | 1997 | -.162*** | -.274*** |
| COVID Knowledge | .286 | 89.128 | 1998 | .269*** | .034 |
| **Other Receptivity-Related Beliefs** | | | | | |
| General interpersonal compassion | .359 | 36.753*** | 496 | .134** | .275*** |
| Concern for others' COVID vulnerability | .501 | 83.228*** | 496 | .122** | .432*** |
| Disgust Sensitivity | .151 | 5.888** | 502 | -.020 | .159** |
| Perceived vulnerability to disease | .227 | 13.632*** | 502 | .116* | .150** |
| Political ideology (higher, more conservative) | .228 | 54.526*** | 1996 | -.183*** | -.075** |
| Belief in conspiracy theories | .257 | 17.522*** | 496 | -.249*** | -.016 |
| **Other Target Characteristics** | | | | | |
| Age | .166 | 28.393*** | 1996 | .137*** | .051* |
| Gender (1 = male/0 = female)[b] | .126 | 16.018*** | 1984 | -.107*** | -.034 |
| Science Literacy | .198 | 10.026*** | 494 | .178*** | .038 |
| Fox News[c] | .177 | 7.650** | 475 | -.190*** | .033 |
| NPR[c] | .118 | 3.325* | 475 | .079 | .058 |
| Papers, Magazines[c] | .141 | 4.790** | 475 | .093 | .071 |

[a] Higher numbers reflect more social distancing behavior.

[b] Participants who responded "other" or "prefer not to answer" were excluded from the analysis.

coded 0 = neither watch last week, nor primary news source, 1 = watched last week or primary, 2 = both

*p < .05

**p < .01

***p < .001

variables also correlated with assessments of the pandemic itself, including more support for the social distancing guidelines, greater concern about the spread of COVID-19, stronger beliefs that the threat posed by the virus had not been exaggerated, and a view that public safety should be prioritized over economic recovery.

*Faith in the government officials* was more complex, just as we anticipated. Whereas greater trust that the State Governors can lead us effectively through the COVID-19 crisis was positively associated with behavioral compliance, the relations were negative when participants considered either President Trump specifically or the federal government more generally. More confidence in those sources was associated with *less* social distancing. Interestingly,

**Table 2. Correlation matrix.**

| | A | B | C | D | E | F | G | H | I | J | K | L | M | N | O | P | Q | R | S | T | U | V | W | X | Y | Z | AA |
|---|---|---|---|---|---|---|---|---|---|---|---|---|---|---|---|---|---|---|---|---|---|---|---|---|---|---|---|
| A. Belief in Value of Science | | | | | | | | | | | | | | | | | | | | | | | | | | | |
| B. Trust in Scientists | .289** | | | | | | | | | | | | | | | | | | | | | | | | | | |
| C. Trust Trump re COVID-19 | -.105* | -.528** | | | | | | | | | | | | | | | | | | | | | | | | | |
| D. Trust Governors re COVID-19 | .185** | .007 | .311** | | | | | | | | | | | | | | | | | | | | | | | | |
| E. Confidence in Federal Gov't | .040 | -.395** | .767** | .423** | | | | | | | | | | | | | | | | | | | | | | | |
| F. General Confidence in Trump | -.101* | -.536** | .927** | .289** | .760** | | | | | | | | | | | | | | | | | | | | | | |
| G. Support Social Distancing | .342** | .366** | -.195** | .364** | -.016 | -.164** | | | | | | | | | | | | | | | | | | | | | |
| H. Worry Contracting Virus | .256** | -.113 | .040 | .180** | .124** | .071 | .330** | | | | | | | | | | | | | | | | | | | | |
| I. Likely to Contract Virus | .169** | -.232** | .074 | .138** | .099* | .093* | .148** | .673** | | | | | | | | | | | | | | | | | | | |
| J. Concern about Spread of Virus | .198** | .212** | -.098* | .226** | -.066 | -.107* | .396** | .387** | .205** | | | | | | | | | | | | | | | | | | |
| K. Threat (not) exaggerated | .113* | .587** | -.495** | .024 | -.374** | -.528** | .456** | .082* | -.054* | .290** | | | | | | | | | | | | | | | | | |
| L. Economy versus Safety | -.209** | -.217** | .286** | -.134** | .122** | .251** | -.479** | -.320** | -.191** | -.267** | -.338** | | | | | | | | | | | | | | | | |
| M. COVID Knowledge | -.007 | .644** | -.464** | -.141** | -.426** | -.487** | .154** | -.300** | -.368** | .083* | .547** | -.019 | | | | | | | | | | | | | | | |
| N. Interpersonal compassion | | | | | | | .308** | -.029 | -.087 | .170** | .327** | -.171** | .370** | | | | | | | | | | | | | | |
| O. Concern for others | | | | | | | .481** | .307** | .191** | .274** | .221** | -.259** | .086 | .483** | | | | | | | | | | | | | |
| P. Disgust Sensitivity | | | | | | | .159** | .347** | .244** | .127** | .119* | -.069 | .010 | | | | | | | | | | | | | | |
| Q. Vulnerability to disease | | | | | | | .213** | .335** | .225** | .190** | -.175** | -.108* | -.390** | | | .356** | | | | | | | | | | | |
| R. Politically conservative | -.300** | -.375** | .522** | .045 | .376** | .522** | -.233** | -.053 | -.011 | -.144** | -.388** | .284** | -.276** | -.184** | -.171** | .170** | -.043 | | | | | | | | | | |
| S. Belief in conspiracy theories | | | | | | | -.136** | .186** | .116** | -.002 | -.376** | .035 | -.571** | | | | | .128** | | | | | | | | | |
| T. Age | -.082 | .107* | -.007 | .039 | -.074 | .017 | .055* | -.075** | -.131** | .096** | .069** | .046* | .164** | .127** | -.028 | -.036 | -.110* | .097* | -.218** | | | | | | | | |
| U. Gender (1 = male/ 0 = female) | .141** | -.119** | .091* | .102* | .134** | .088 | -.056 | -.018 | .041 | -.086* | -.140** | -.007 | -.182** | -.221** | -.167** | -.009 | -.087 | .036 | .084 | -.117** | | | | | | | |
| V. Area Re-opened (1 = Yes/ 0 = No) | .054 | -.124** | .120** | -.048 | .141** | .120** | -.044 | .049* | .116** | -.015 | -.131** | .047* | -.204** | -.050 | -.026 | .115** | .041 | .093* | .228** | -.007 | .022 | | | | | | |
| W. Science Literacy | .007 | .514** | -.396** | -.125** | -.436** | -.409** | .021 | -.260** | -.267** | .030 | .348** | .089* | .621** | | | | | -.211** | | .212** | -.079 | -.170** | | | | | |
| X. Fox News[b] | | | | | | | -.168** | -.032 | -.015 | -.053 | -.275** | .191** | -.149** | | | | | .412** | .151** | .126** | -.007 | .112* | | | | | |

*(Continued)*

**Table 2.** (Continued)

| | A | B | C | D | E | F | G | H | I | J | K | L | M | N | O | P | Q | R | S | T | U | V | W | X | Y | Z | AA |
|---|---|---|---|---|---|---|---|---|---|---|---|---|---|---|---|---|---|---|---|---|---|---|---|---|---|---|---|
| Y. NPR[b] | | | | | | | .157** | .000 | .090* | .071 | .162** | -.122** | .114* | | | | | -.221** | -.032 | -.025 | .057 | -.087 | | -.196** | | | |
| Z. Papers, Magazines[b] | | | | | | | .124** | .006 | .047 | .106* | .167** | -.053 | .210** | | | | | -.210** | -.220** | .035 | -.007 | -.092* | | -.191** | .124** | | |
| AA. Self-Report Distancing | .223** | .257** | -.096* | .267** | .042 | -.095* | .611** | .275** | .093** | .359** | .342** | -.348** | .158** | .339** | .490** | .150** | .202** | -.159** | -.137** | .114** | -.083** | -.052* | .115* | -.057 | .095* | .114* | |
| BB. Behavioral Distancing | .192** | .354** | -.292** | .138** | -.162** | -.279** | .476** | .241** | .064** | .339** | .428** | -.287** | .284** | .266** | .328** | .051 | .183** | -.218** | -.256** | .160** | -.122** | -.095** | .195** | -.174** | .106* | .126** | .459** |

\*\* $p < .01$

\* $p < .05$; Cells without an entry involve variables that were included in different sub-studies.

these relations do not appear to be a simple reflection of political orientation. Although participants who more strongly identified as conservative engaged in less social distancing and expressed more trust/confidence in President Trump, in each case the measures of social distancing accounted for unique variance over and above that explained by political orientation (all $p$'s $< .001$).

All the *belief measures concerning the pandemic itself* related as expected with social distancing. This was especially true of support for the social distancing guideline, worry about contracting the virus, concern about the spread of the virus, and the assessment that the threat posed by the virus had not been exaggerated. Believing that relatively more emphasis should be placed on economic recovery than public safety also was associated with less social distancing.

Answers to our test of *COVID-19 knowledge* also related positively to behavioral compliance. Importantly, the recognition of true statements and the rejection of misinformation each correlated with social distancing (multiple $R$'s of .234 and .260, respectively). Knowledge also was associated with support for the social distancing guideline and especially with the belief that the threat posed by the coronavirus had not been exaggerated. In addition, more knowledgeable individuals expressed greater trust in scientists and less confidence in President Trump.

Self-beliefs highlighting *interpersonal compassion* and *concern for others' vulnerability* to the virus were associated with more social distancing. These variables correlated as expected with beliefs about the pandemic. For example, more compassionate individuals were more supportive of the social distancing guideline and believed that the threat of the virus had not been exaggerated. The same was true of individuals who had expressed concern for others' vulnerability; they also were more worried that they themselves would contract the virus.

The data offered a number of interesting observations regarding the extent to which respondents viewed themselves as generally *vulnerable to disease*. This variable was related to more social distancing, and, just as one would expect, with greater worry about contracting COVID-19 and greater perceived likelihood of contracting it. Disgust sensitivity correlated with perceived vulnerability to disease, replicating past findings, and also related to social distancing. Stronger disgust sensitivity also was associated with greater worry about contracting COVID-19 and greater likelihood of doing so.

As already noted, *political orientation* also was relevant; more conservative individuals engaged in less physical distancing. Just as expected, political ideology correlated strongly with general confidence in President Trump and trust in his leadership regarding the COVID-19 crisis, but not with trust in the state Governors. More conservative individuals also reported less belief in the value of science and less trust in scientists. They also believed the threat of the coronavirus to have been exaggerated and that economic considerations needed to take priority over public safety.

Generally believing in *conspiracy theories* also was predictive of less social distancing, possibly because it promotes a less accurate view of the pandemic. Indeed, such beliefs correlated strongly with scores on the test of COVID-19 knowledge. Conspiracy theorists also were more likely to believe that the threat posed by the coronavirus had been exaggerated.

In addition to beliefs, we examined a number of other personal characteristics that seemed potentially relevant to receptivity to the directive (see the fourth section of Table 1). Female participants displayed more evidence of social distancing, as did older participants. Our hypothesis regarding *science literacy* also received support. Those who exhibited a greater understanding of a small set of basic scientific facts engaged in more social distancing. Scientific literacy also related strongly to expressed trust in scientists and scores on the test of COVID-19 knowledge. It also was associated with the belief that the threat posed by the virus had not been exaggerated.

Finally, although none of the multiple correlations were very substantial, accounting for less than 3% of the variance, a number of the *news sources* variables related to social distancing. Whereas engagement with NPR or newspapers and magazines was associated positively social distancing, greater involvement with Fox News related negatively to distancing. The latter was more common for participants who endorsed a more conservative political orientation, whereas the former was associated more strongly with a more liberal perspective. Parallel relations were observed with support for the social distancing guideline, valuing economic considerations more than public safety, believing that the COVID-19 threat had been exaggerated, and accurate knowledge regarding the virus.

## Comparing the virtual behavior and self-report measures of social distancing

Our primary interest was to employ the virtual behavior and self-report measures of social distancing as supplemental to one another and, hence, as jointly related to each of the hypothesized predictor variables. Although the two measures were related ($r = .459$, $p < .001$), the magnitude of the correlation was not so overwhelming as to suggest that they were equivalent. Given this observation, it is interesting to consider how the two variables differ with respect to the unique variance for which they each accounted in the various multiple regressions reported in Table 1. For COVID-19 knowledge and scientific literacy, the multiple correlation was driven almost entirely by the virtual behavior measure. Indeed, the self-report failed to account for any significant unique variance. The same was true with respect to engagement with Fox News, belief in conspiracy theories, and the variables reflecting trust in President Trump's leadership regarding the pandemic and general confidence in him. On the other hand, for the measure of support for the social distancing guideline, the self-report measure of social distancing accounted for twice the unique variance that was associated with the virtual behavior measure. Similar patterns were evident for general compassion, concern for others' vulnerability to COVID-19, and disgust sensitivity.

In order to statistically compare the relations with the virtual behavior measure to those with the self-report measure, we tested the difference between each pair of simple correlations with a *t*-test for dependent correlations [51]. Table 3 summarizes this comparison. Any predictor variable for which the comparison yielded *p*-value less than .05 is listed. Although drawing

**Table 3.  Comparing virtual behavior and self-report measures of social distancing.**

| | Behavioral *r* | Self-Reported *r* | *t*-test of Difference | *p* = |
|---|---|---|---|---|
| **Beliefs about the Source** | | | | |
| Trust President Trump re COVID-19 crisis | -.292 | -.096 | 3.206 | .002 |
| Trust State Governors re COVID-19 crisis | .138 | .267 | -2.179 | .030 |
| Confidence in Federal Gov't General Effectiveness | -.162 | .042 | 3.242 | .001 |
| General Confidence in President Trump | -.279 | -.095 | 3.019 | .003 |
| **Beliefs about the Context** | | | | |
| Support Social Distancing Guideline | .476 | .611 | -5.870 | .000 |
| Threat (not) exaggerated | .428 | .342 | 3.587 | .000 |
| Economy More Important than Safety | -.287 | -.348 | -2.412 | .016 |
| COVID Knowledge | .285 | .158 | 5.007 | .000 |
| **Other Receptivity-Related Beliefs** | | | | |
| Concern for others' COVID vulnerability | .386 | .490 | -2.066 | .039 |
| Political ideology (higher #, more conservative) | -.218 | -.159 | 2.090 | .037 |
| Belief in conspiracy theories | -.256 | -.137 | 2.081 | .038 |

any strong inferences from these patterns is difficult, the two statistically largest differences appear especially striking and accord well with the above observations concerning unique variance. COVID-19 knowledge, which is a much more objective measure than any of the other predictor variables, correlated more strongly with the virtual behavior measure. On the other hand, the measure that is arguably most subjective–support or opposition for the social distancing guideline–correlated much more strongly with the self-report measure of social distancing.

It appears that a sizeable number of participants may have offered self-reports that were overestimates of their actual social distancing behavior. Whereas the virtual behavior data displayed a normal distribution, the distribution of scores on the self-report measure was skewed with a substantial majority responding at or near the positive endpoint of the scale (M = 5.98 on a 7-point scale, SD = 1.18). Such overestimation is to be expected to the extent that participants wished to believe themselves as having acted in manners that avoided placing their health, or that of others, at risk, or were simply concerned with responding in socially-desirable fashion. As noted earlier, such self-beliefs and concerns have been shown to influence retrospective memory processes [21, 25, 26].

To examine such overestimation more systematically, we focused on the residuals from a simple regression predicting scores on the self-report measure of social distancing from the virtual behavior measure. More positive residuals reflect a self-report score that is higher than expected on the basis of the social distancing exhibited on the virtual behavior items. We then correlated these residuals with each of our predictor variables, in order to assess the extent to which each related to such statistical overestimation. Table 4 lists any predictor variable for

**Table 4. Correlations with the residual predicting self-report measure of social distancing from the virtual behavior measure[a].**

| | r | p | n |
|---|---|---|---|
| **Beliefs about the Source** | | | |
| Belief in Value of Science | .151 | .001 | 497 |
| Trust in Scientists | .107 | .017 | 497 |
| Trust State Governors re COVID-19 crisis | .226 | .000 | 496 |
| Confidence in Federal Gov't Effectiveness | .129 | .004 | 496 |
| **Beliefs about the Context** | | | |
| Support Social Distancing Guideline | .442 | .000 | 1998 |
| Worry about Contracting Virus | .185 | .000 | 2001 |
| Likely to Contract Virus | .072 | .001 | 2000 |
| Concerned about the Spread of the Virus | .233 | .000 | 2000 |
| Threat (not) exaggerated | .164 | .000 | 2001 |
| Economy More Important than Safety | -.244 | .000 | 2000 |
| **Other Receptivity-Related Beliefs** | | | |
| General interpersonal compassion | .237 | .000 | 499 |
| Concern for others' COVID vulnerability | .374 | .000 | 499 |
| Disgust Sensitivity | .143 | .001 | 505 |
| Perceived vulnerability to disease | .136 | .002 | 505 |
| Political ideology (higher, more conservative) | -.066 | .003 | 1999 |
| **Other Target Characteristics** | | | |
| Age | .045 | .043 | 1999 |

[a] Higher numbers reflect greater reports of social distancing than expected on the basis of the virtual behavior measure.

[b] coded 0 = neither watch last week, nor primary news source, 1 = watched last week or primary, 2 = both

which the correlation was statistically significant. In general, the more participants held beliefs associated with a serious view of the pandemic (e.g., greater belief in science, worry about the coronavirus, concerns about their own and others' vulnerability), the more their self-reports of social distancing were greater than expected on the basis of their virtual distancing behavior. Especially noteworthy, once again, is the extent to which participants supported or opposed the social distancing guideline, for which the correlation with the residual was the highest of any variable. The more support participants expressed, the more their retrospective reports of compliance with the social distancing directive appeared to be overestimates. These more supportive individuals reported having followed the social distancing recommendations to a much greater extent than would be expected on the basis of their "in-the-moment" decisions on the graphical scenario items that comprised the virtual behavior measure of social distancing.

## Discussion

The findings highlight the importance of individuals' beliefs as factors associated with social distancing behavior. They also support the theoretical framework of compliance that guided our selection of variables for inclusion in the study. Any directive regarding behavior change will be shaped by beliefs about the directive's source, beliefs about the context surrounding the challenge to which the directive is responding, and relevant self-views and characteristics. As such, the framework is applicable to any call for behavior change aimed at the general public. When applied to the specific challenge posed by the spread of the COVID-19 virus and the directive to engage in social distancing, the conceptual framework led to our focus on (a) source variables related to the government and public health officials, (b) beliefs regarding COVID-19 and the severity of the threat it posed, and (c) various self-related beliefs and target characteristics influencing receptivity to the social distancing directive.

Importantly, these relations were evident not only on a self-report social-distancing measure but also on a measure that relied on vivid, graphical simulations of real-life behavior. Participants made concrete, "in-the-moment" decisions about actions involving different degrees of social distancing. They interactively distanced themselves from oncoming passersby, from people standing in line, and from fellow grocery shoppers, coffeeshop customers, and library patrons. They selected a position on a crowded beach and traversed a crowded plaza. As such, the behavioral decisions, albeit virtual, closely matched the features of real-life situations.

The current findings did indeed reveal some striking differences between behavioral and self-report measures of social distancing. Although the two were related, the correlation did not reach a level that suggested these were equivalent measures of the same construct. Moreover, both the simple correlations and the unique variance accounted for by each measure differed markedly for a number of predictor variables. Especially telling was that scores on our tests of COVID-19 knowledge–the most objective of our predictor variables–related more strongly to the behavioral measures, with self-reports accounting for little or no additional variance. Thus, self-reports do not cohere with behavioral decisions sufficiently to suggest they are mutually interchangeable. The findings suggest that retrospective reports of social distancing behavior may be unduly influenced by attitudes toward social distancing guidelines and self-beliefs that imply the desired behavior. Nevertheless, the virtual behavior measure of social distancing and the self-report measure did complement one another well, as is evident by their accounting for unique variance for many predictor variables.

Before concluding, we do wish to acknowledge a few important limitations regarding the present research. First, the participants were U. S. residents and many of the survey items referenced that particular context. Hence, whether and to what degree these findings can be

generalized to other nations, especially ones that managed to avoid politicizing the pandemic to the extent that has been true in the United States. Second, we made a strategic methodological decision to use a "planned missing" design, segmenting the various predictor variables into four subsets, to which the participants were randomly assigned. This allowed us to keep the survey for each individual participant relatively brief and, hence, lessen the possibility of participant fatigue and associated confounds. However, it did come at a cost. This design means that we are unable to assess the relations between some of the predictors and are unable to empirically confirm (e.g., via factor analysis) the categorical distinctions that comprise our guiding theoretical framework: beliefs about the source of the directive, beliefs about the challenge the directive addresses, and relevant target characteristics. Future research may address this limitation through the use of multi-wave surveys or other means of countering attrition and inattentive responding in longer-format surveys. With responses to each measure from every participant, a factor analysis could examine the extent to which our conceptual categorizations are supported by the data. Finally, although we assessed participants' knowledge regarding COVID-19 with a series of true and false statements regarding its spread and treatment, we did not specifically address understanding of the social distancing recommendations. Any participants who were either unaware of the guidelines or misunderstand them are unlikely to have behaved accordingly. However, we suspect that any such misunderstandings would correlate strongly with scores on our test of COVID-19 knowledge.

We conclude with a brief consideration of the implications of the present findings for public service campaigns encouraging social distancing. How might compliance be promoted? The literature regarding scientific communications (e.g., those concerning climate change, vaccinations, or stem-cell research) highlights that persuading individuals to adopt scientifically-sound beliefs and modify their behavior accordingly is fraught with difficulties, especially as an issue becomes politicized [52]. Message recipients often fail to process information accurately. Various motivated reasoning processes, including source derogation, counterarguing, and sheer denial, allow individuals who are exposed to a counterattitudinal message to reach a desired conclusion, thus failing to disconfirm, and sometimes lending support, to their preexisting beliefs and ideology [53–55]. Given this critical barrier to effective science communication, many researchers have emphasized the importance of attending to the motivations that underlie science-skeptical attitudes [56, 57] and the value of tailoring messages to the audience such that functionally equivalent information is framed in a manner that is consistent with ideological values [58–60].

Unfortunately, the pandemic has become extraordinarily politicized within the U.S., much more so than in such countries as Canada, Germany, and South Korea whose leaders pursued a more consistent and pragmatic approach to the initial wave of the pandemic [61–64]. That politicization is very evident in the present data. Very different relations were observed with respect to trust in President Trump versus the State Governors as providing effective leadership during the early months of the COVID-19 crisis. Participants' political orientation, and even their exposure to more partisan news sources, related to beliefs about the severity of the COVID-19 threat, support for the social distancing guideline, and social distancing behavior.

It is precisely this politicization that poses such a barrier to effective science communication regarding the pandemic. Message tailoring surely will be critical with respect to promoting acceptance of and compliance with social distancing recommendations. Campaigns are more likely to be effective when they address the motivational roots underlying minimization of the severity of the pandemic and accord with individuals' social identity needs [57, 65–67]. Multiple strategies are likely necessary for widespread acceptance. Although the present findings are silent with respect to the strategies themselves, they do offer some insights regarding the content that should be emphasized, albeit framed optimally. The findings highlight the importance

of communicating accurate knowledge, as well as dispelling misinformation, about COVID-19 (how it spreads and how the risks of contraction can be mitigated). There is also likely value in appealing to, and heightening concern, about others' vulnerability to the coronavirus and the suffering of those infected. Similarly, the results regarding perceived vulnerability to disease and compassion suggest the need to emphasize the vulnerability of people of all ages to the virus and the role that everyone, whether symptomatic or not, plays in spreading it. Indeed, the data seem to call for frequent repetition of the portrayal of social distancing guidelines that White House coronavirus response coordinator Dr. Deborah Birx offered: "This is a road map to prevent your grandmother from getting sick" [68].

The virtual behavior measures of social distancing that we employed in this research also may prove helpful in the context of public health campaigns. They certainly could be used as educational tools to illustrate appropriate social distancing behavior. Moreover, following some educational intervention, they could serve as exercises that encourage individuals to rehearse behaviors that abide by social distancing recommendations. The virtual behavior items also could be employed, sometime after exposure to an intervention, as outcome measures to test the effectiveness of a persuasive campaign.

## Supporting information

**S1 Material. Description of the simulated behavioral measures of social distancing.** (PDF)

**S2 Material. Demographic details for each of the four sub-studies.** (PDF)

## Author Contributions

**Conceptualization:** Russell H. Fazio, Benjamin C. Ruisch, Courtney A. Moore, Javier A. Granados Samayoa, Shelby T. Boggs, Jesse T. Ladanyi.

**Data curation:** Benjamin C. Ruisch.

**Funding acquisition:** Russell H. Fazio.

**Methodology:** Russell H. Fazio, Benjamin C. Ruisch, Courtney A. Moore, Javier A. Granados Samayoa, Shelby T. Boggs, Jesse T. Ladanyi.

**Project administration:** Russell H. Fazio.

**Resources:** Benjamin C. Ruisch, Courtney A. Moore.

**Software:** Benjamin C. Ruisch.

**Supervision:** Russell H. Fazio.

**Writing – original draft:** Russell H. Fazio.

**Writing – review & editing:** Russell H. Fazio, Benjamin C. Ruisch, Courtney A. Moore, Javier A. Granados Samayoa, Shelby T. Boggs, Jesse T. Ladanyi.

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
