## [Decision Letter · Decision Letter 0]

11 Nov 2020

PONE-D-20-30793

Who Is (Not) Complying with the Social Distancing Directive and Why?  Testing a General Framework of Compliance with Multiple Measures of Social Distancing

PLOS ONE

Dear Prof. Fazio,

Thank you for submitting your manuscript to PLOS ONE. After careful consideration, we feel that it has merit but does not fully meet PLOS ONE’s publication criteria as it currently stands. Therefore, we invite you to submit a revised version of the manuscript that addresses the points raised during the review process.

Your manuscript was carefully reviewed by two expert social psychologists with a track record of research in attitude-behaviour relationships and applied social and health psychology. One of the reviewers suggested minor revision, while the second suggested major revision. I concur with the second reviewer (major revision required) and kindly ask you to consider the points raised and provide a point-by-point response letter, should you decide to revise and resubmit your work.

We look forward to receiving your revised manuscript.

Kind regards,

Lambros Lazuras

Academic Editor

PLOS ONE

Journal Requirements:

Reviewers' comments:

Reviewer's Responses to Questions

**Comments to the Author**

1. Is the manuscript technically sound, and do the data support the conclusions?

Reviewer #1: Yes

Reviewer #2: Partly

2. Has the statistical analysis been performed appropriately and rigorously? 

Reviewer #1: Yes

Reviewer #2: Yes

3. Have the authors made all data underlying the findings in their manuscript fully available?

Reviewer #1: Yes

Reviewer #2: Yes

4. Is the manuscript presented in an intelligible fashion and written in standard English?

Reviewer #1: Yes

Reviewer #2: Yes

5. Review Comments to the Author

Reviewer #1: This paper tests a theoretical model of compliance with social directives in the context of social distancing behaviors relevant to COVID-19 infection. The research is novel in testing a new model and comprehensive in testing multiple predictors of both self-reported and objectively measured behavior. The model is assessed using a cross-sectional survey of 2,000 MTurk workers. The data, analysis, and write-up are all of a high standard. Overall, I am very favorably disposed towards publication of this research in PLoS One.

There are a small number of issues that the authors might wish to consider in revising the manuscript.

1. I found theoretical framework compelling and liked the division of predictors into beliefs about the context, beliefs about the source, and target characteristics. I can see the conceptual basis of the distinctions and wondered if this categorization can also be supported empirically (e.g., via factor or cluster analysis)?

2. The behavioral measure of social distancing is very clever and adds an interesting dimension to the research. At the same time, is it accurate to suggest that it is an “objective” measure of “behavior”? The scenarios are hypothetical, and participants are asked about their comfort levels within each one. Is this more a measure of willingness (or something else) than behavior? If so, would it be worthwhile to predict discrepancies between willingness and self-reported behavior?

3. The differences between the factors that predict “objective” and self-report measures of behavior are intriguing. Could the analyses go further and determine whether these seeming differences are significant? For instance, could MLM be used here with the two outcomes as a within-participants factor?

4. Table 1 offers a long list of significant predictors of social distancing (attesting to the model’s value) and I appreciated the multivariable analyses of predictors within each set of beliefs. Would the research benefit from an additional analysis that included all 19 beliefs in the model to determine which beliefs or characteristics are most important, and how much variance in the outcomes they collectively explain?

Reviewer #2: The authors present an article that reports upon the findings of an online, survey-based study designed to identify more about the antecedents of people’s willingness to comply with social distancing measures in the wake of the COVID-19 pandemic. Two distinct DVs are used; self-report measures and a novel ‘behavioral’ measure.

The research is timely, given the unfolding nature of the global pandemic, and there are also potential theoretical and applied implications for the work. However, there are a number of significant issues with the study – as reported – that I feel preclude publication on PLOS One at this time.

A key issue is the lack of detail provided in each of the sections of the study (particularly the introduction, methods and discussion). There needed to be more methodological detail to allow for the replication of this study, if necessary. In terms of the results, I think that the basic analysis is okay, but I wonder whether the authors could be more ambitious with their modelling. The discussion is also notably light on detail – more effort needed to be made to couch the findings of the study among those from the rich, extant literature on science communication.

It is possible that with major corrections/additions that this article can be shaped into a publishable offering. What follows is a specific appraisal of the key issues with the article, from my perspective. I wish the authors all the best in making the suggested amendments and will look forward to appraising the article in due course.

Title

It should be stated within the title that the focus is on compliance within the US. The study is clearly tied to social distancing directives in the US and the study is conducted on a US sample. The current title infers that the scope of the study transcends the US context, which is not the case.

Introduction.

The introduction would benefit from a short paragraph outlining the COVID-19 crisis and stating more about the situation and response in the US. Again, this is important to clarify the context and reach of the study, which is US-centric. This US-centricity is exemplified by the claim that “stay six feet away from others” is a regularly repeated mantra. This might be the case for the US, but this is not the case in Europe, for example, where the mantra has been more ‘metric’ (2 meters or 1 meter with extra precautions).

Line 35 – I feel that the claim that people will only respond to directives that are justified needs to be more nuanced. There are examples of where people comply with directives that are not deemed to be well-justified, because they do not wish to suffer social/group disapproval.

Line 41 – Please outline a relevant (e.g. health-related) example of a study that evidences the point you are making here.

Line 44 – You have evidenced the source and audience as being important, but what about message factors? If you are treating message factors as part of the ‘surrounding context’ then I feel that this needs to be explicitly stated. If you are not classing message factors as important, then this needs to be strongly justified, dismissing the ‘what’ from the ‘who is saying what to whom?’ mantra would otherwise appear to be a significant oversight.

Line 49-50 – I think that the phrasing here needs to be more nuanced. It is not always going to be the case that people who process information in a favorable way will engage in the correct/sustained/sufficient behaviors (we see this a lot with health and environmental actions, where people often engage in compensatory behavioral trade-offs). Equally, sometimes people who are less positive about the directions of a message, might still comply with them.

Line 58 – Please provide a reasoned example of the characteristics/beliefs that might affect mis-representation of social distancing on self-report measures.

Line 66 – I need more convincing that the novel behavioral measures you have designed have the ecological validity that you are claiming. Granted they do present people with in the moment behavioral decisions, but these are virtual and vicarious. Please provide a fuller justification of the validity of your ‘graphical scenarios’ in emulating real-life decision making.

Line 75-112 – There needs to be a fuller, theoretical justification for the selection of the various predictors used in this study (source, context and target). While I am not disputing the relevance of the predictors you have selected, the selection process is opaque, and the selection of items appears subjective and unsystematic. For example, your decision to look at ‘objective knowledge of COVID-19’ as opposed to ‘subjective knowledge’ is interesting, given that subjective knowledge of science/technological issues often shares a stronger relationship with acceptance (see, e.g., some of the work by Stugis and Allum).

Tying the selection of variable more explicitly to an established over-arching theoretical framework(s) would have made for a more compelling narrative. It would have also allowed you to be more adventurous in modeling moderation/mediation pathways.

Line 88 – You need to provide evidence for the claim that Trump downplayed the severity relative to state Governors (I believe you, but you need evidence).

Methods

Line 123 – Strictly speaking ‘MTurkers’ is not a word and so you should probably put this is ‘inverted commas’

Line 127 – I understand the arguments for creating four versions of the survey, but you should probably raise the fact that participants did not all complete the same survey as a limitation to the study within the discussion.

Line 131 – Bearing in mind you have a between-subjects design, you need to show evidence that there has been successful randomization to condition in terms of core participant demographics. Please provide some evidence of this – means/frequencies, plus basic statistical comparative analysis will be sufficient.

Line 136 – Related to my point above, you need to have specified that there was a national lockdown in the US (and the nature of this lockdown), rather than assuming that the readership of the article will know/be aware of this.

Line 140 – Please specifically refer to COVID-19 as there could be other COVIDs in the future (although we hope not of course!)

Line 144 – What kind of selection, piloting and pre-testing did your social distancing behavioral measures go through? Now, they might have a degree of face-validity as measures of social-distancing compliance, but what formal evidence is there of their content-validity beyond this? While I do applaud the innovation in measurement here, I am concerned that that drawing strong inferences based on an un-proven, un-tested measure is ill-advised. As such you need to provide a fuller account of the selection and development of these measures, while also acknowledging the limitations around their uses within the discussion.

Line 156-233 – You need to include the responses options for all your measures. You also need to provide access (probably in the form of an appendix or supplementary information) to your questionnaire measures, so people can see all the questions that were asked and how they were asked. Within the text you might wish to add two example questions/statements per scale outlined.

Line 160 – Did you assess people’s awareness and understanding of the social distancing recommendations? If not, this is a limitation that needs consideration within the discussion.

Line 168 – How many statements were used to assess COVID-19 knowledge?

Line 199 – For those outside the US it would help to know what political leaning each of these news/media sources have.

Line 202 – Revisit this sentence and see if you can rephrase it to make it clearer.

Line 226 – What demographics were included and how were they assessed?

Results

The results appear to be okay but very basic given the scope of the study and the data you have to work with. I wonder if you could have been more ambitious with your modeling/analysis, to say something a little more about who is responding to COVID-19 guidance or not?

This point of critique maps back to a point I made early re: the selection of the predictors, which – although not disputing their relevance - appears quite haphazard. Your narrative would be substantially more compelling if you were able to chart, model and report some moderation/mediation pathways. You have a considerable number of predictors here, which should make it possible, provided that theoretically justified pathways can be created.

Line 308 – Please state ‘COVID-19’.

Discussion

The discussion needs a lot of work. Lines 328-351 essentially just re-report detail that has just been outlined within the results section. As such, this detail can be pared back somewhat.

The discussion from Line 352 onwards is basic fails to map clearly and robustly to the rich literature that exists around science communication. The claims that are being made, need to be more couched within the extant literature and explicitly linked back to the study findings. There also needs to be a more upfront consideration of the limitations of the research.

6. PLOS authors have the option to publish the peer review history of their article (what does this mean?). If published, this will include your full peer review and any attached files.

Reviewer #1: No

Reviewer #2: No

---

## [Author Response · Author response to Decision Letter 0]

12 Dec 2020

See the uploaded file "Response to Reviewers."

---

## [Decision Letter · Decision Letter 1]

18 Jan 2021

PONE-D-20-30793R1

Who Is (Not) Complying with the U. S. Social Distancing Directive and Why?  Testing a General Framework of Compliance with Multiple Measures of Social Distancing

PLOS ONE

Dear Dr. Fazio,

Thank you for submitting your revised manuscript to PLOS ONE. Both reviewers expressed their satisfaction with the revisions made, and almost all of their comments and suggestions have been accommodated. Your manuscript can be accepted for publication subject to addressing a few minor issues identified by the second reviewer. Addressing those issues will further strengthen your manuscript, and I am hoping that you are willing to consider them.

We look forward to receiving your revised manuscript.

Kind regards,

Lambros Lazuras

Academic Editor

PLOS ONE

Reviewers' comments:

Reviewer's Responses to Questions

**Comments to the Author**

1. If the authors have adequately addressed your comments raised in a previous round of review and you feel that this manuscript is now acceptable for publication, you may indicate that here to bypass the “Comments to the Author” section, enter your conflict of interest statement in the “Confidential to Editor” section, and submit your "Accept" recommendation.

Reviewer #1: All comments have been addressed

Reviewer #2: (No Response)

2. Is the manuscript technically sound, and do the data support the conclusions?

Reviewer #1: Yes

Reviewer #2: Yes

3. Has the statistical analysis been performed appropriately and rigorously? 

Reviewer #1: Yes

Reviewer #2: Yes

4. Have the authors made all data underlying the findings in their manuscript fully available?

Reviewer #1: Yes

Reviewer #2: Yes

5. Is the manuscript presented in an intelligible fashion and written in standard English?

Reviewer #1: Yes

Reviewer #2: Yes

6. Review Comments to the Author

Reviewer #1: I had no substantive concerns about the original version of this manuscript and merely made some suggestions that I hoped might serve to increase the impact of an already compelling and important piece of research. The authors were very responsive to my suggestions and the changes to the manuscript made a very good paper even better. I am very happy to recommend publication of this research in PLoS ONE.

Reviewer #2: I thank the authors for taking the time to respond to my previous comments so thoroughly. I feel that the article is now much improved. I do have a few additional minor points that I feel should be considered before this article can be accepted for publication. I outline these points below. The page numbers that area identified relate to the revised version of the manuscript that includes the highlighted additions, changes and omissions.

Title, Abstract and Introduction

The title, abstract and introduction are now much improved, however, I have a couple of final suggestions for these sections:

1. I feel that the title would benefit from mentioning COVID-19 – as the work focuses on compliance with social distancing requests in response to this specific pandemic.

2. On page 2-3, it might be beneficial to incorporate a couple of photographs to depict what the social measures (e.g. tape on the floor) look like. This should hopefully be simple to get hold of as University campuses tend to have such measures (if they are accessible during the pandemic)

Methods

The methods section is now fuller. Thank you for including more of the detail, rather than placing this all in the supplementary material. It might add to the article if you were to include a couple of ‘stills’ depicting the virtual behaviour measures. I understand these are in the ‘Supplementary Material’ but having a couple of images in the main text would be good for to illustrate the innovative method you have used.

Results

Looking at Table 1, I am wondering how you coded the people who did not enter their gender? It could be that these individuals wished to withhold information about their gender, but equally it could be that these individuals identify as non-binary. In the spirit of inclusivity, you should probably add a footnote to Table 1 to note that people withholding gender were not included.

Overall, while I find the results section relatively simple bearing in mind the data-set that has been accrued, I sense that my requirement for more in-depth inferential analysis should be traded-off against the value of getting this article into the public realm (also, I appreciate the issues caused by the methods used). I would encourage you to perhaps outline in the discussion section what the obvious ‘next steps’ might be for the work, which might allow for more in-depth inferential statistics to be performed – one or two suggestions would suffice.

I am wondering whether using a basic t-test to compare the strengths of the correlation coefficients is appropriate (see Table 3). This is something that you should check, as I feel that there might be more statistically robust ways of drawing such comparisons. For example, this recent article in PLOS offers up suggestions and techniques for drawing such comparisons: cocor: A Comprehensive Solution for the Statistical Comparison of Correlations (plos.org). If the current analysis is appropriate, then including a couple of references to studies that have used this method of analysis should provide adequate justification for the approach.

P22 - you suggest that “Any predictor variable for which the comparison yielded a significance level greater than .05 is listed”; do you mean ‘lower than .05’?

Also, on P22 you state “…it is interesting to consider how the two variables differ with respect to the unique variance for which they each accounted in the multiple regressions”. The end of this sentence suggests you have done a multiple (linear) regression analysis, when in reality you have done a multitude of regressions (unless I am mistaken). As such, you should probably look to reword this sentence.

P23 - you suggest that the skew in the self-report measure was due to people wishing to believe that they had acted in a way to lessen their and others’ risk. It could also be that people wished to present themselves in such a way. This should probably be accounted for in your statement.

P25 - I find the paragraph beneath the table quite difficult to parse. Is there are way to rephrase things slightly to make it easier to understand.

Discussion

P27 – do you have countries that you can reference where there was less politicisation of the pandemic? You might wish to mention South Korea, which seems to have done this effectively. Also, there are interesting contemporary articles showing how politicisation has affected the COVID response that you might wish to mention (e.g. Crayne and Medeiros, 2020). These articles might also be applicable to some of your commentary on politicisation on P28.

Crayne, M. P., & Medeiros, K. E. (2020). Making sense of crisis: Charismatic, ideological, and pragmatic leadership in response to COVID-19. American Psychologist. Advance online publication. http://dx.doi.org/10.1037/amp0000715

I generally like the additional commentary regarding the implications for the work, however, I wonder if something more could be said. I do not dispute that we should be looking to communicate accurate information and dispel myths, but this conclusion is fairly standard. Is it possible to say something more about how you could (perhaps) adapt your virtual behavior measures as an educational tool (bearing in mind the correlation with COVID-19 risk)? Also, maybe you could comment a bit more concretely about strategies that could be used to overcome the issues you outline (e.g. Hornesy and Fielding, 2017). I don’t think that there needs to be much added here, but a little more specificity around the guidance and suggestions would give the article more ‘bite’. Relatedly, adding in a few more references to back up you claims about how emotional appeals around vulnerability might be effective would be good.

Hornsey, M. J., & Fielding, K. S. (2017). Attitude roots and Jiu Jitsu persuasion: Understanding and overcoming the motivated rejection of science. American Psychologist, 72(5), 459-473. http://dx.doi.org/10.1037/a0040437

7. PLOS authors have the option to publish the peer review history of their article (what does this mean?). If published, this will include your full peer review and any attached files.

Reviewer #1: **Yes: **Paschal Sheeran

Reviewer #2: No

---

## [Author Response · Author response to Decision Letter 1]

5 Feb 2021

See the uploaded file "Response to Reviewers"

---

## [Editor Report · Decision Letter 2]

9 Feb 2021

Who Is (Not) Complying with the U. S. Social Distancing Directive and Why?  Testing a General Framework of Compliance with Virtual Measures of Social Distancing

PONE-D-20-30793R2

Dear Dr. Fazio,

We’re pleased to inform you that your manuscript has been judged scientifically suitable for publication and will be formally accepted for publication once it meets all outstanding technical requirements.

Kind regards,

Lambros Lazuras

Academic Editor

PLOS ONE

---

## [Editor Report · Acceptance letter]

17 Feb 2021

PONE-D-20-30793R2 

Who is (Not) Complying with the U. S. Social Distancing Directive and Why?
Testing a General Framework of Compliance with Virtual Measures of Social Distancing 

Dear Dr. Fazio:

I'm pleased to inform you that your manuscript has been deemed suitable for publication in PLOS ONE. Congratulations! Your manuscript is now with our production department. 

Kind regards, 

on behalf of

Dr. Lambros Lazuras 

Academic Editor

PLOS ONE